# A retrospective study investigating requests for self-citation during open peer review in a general medicine journal

**Erin Peebles, Marissa Scandlyn, Blair R. Hesp** *

Kainic Medical Communications Ltd., Dunedin, New Zealand

* blair@kainicmedical.com

## Abstract

### Introduction

Peer review is a volunteer process for improving the quality of publications by providing objective feedback to authors, but also presents an opportunity for reviewers to seek personal reward by requesting self-citations. Open peer review may reduce the prevalence of self-citation requests and encourage author rebuttal over accession. This study aimed to investigate the prevalence of self-citation requests and their inclusion in manuscripts in a journal with open peer review.

### Methods

Requests for additional references to be included during peer review for articles published between 1 January 2017 and 31 December 2018 in *BMC Medicine* were evaluated. Data extracted included total number of self-citations requested, self-citations that were included in the final published manuscript and manuscripts that included at least one self-citation, and compared with corresponding data on independent citations.

### Results

In total, 932 peer review reports from 373 manuscripts were analysed. At least one additional citation was requested in 25.9% (n = 241) of reports. Self-citation requests were included in 44.4% of reports requesting additional citations (11.5% of all reports). Requests for self-citation were significantly more likely than independent citations to be incorporated in the published manuscript (65.1% vs 52.1%; chi-square p = 0.003). At the manuscript level, when requested, self-citations were incorporated in 76.6% of manuscripts (n = 72; 19.3% of all manuscripts) compared with 68.5% of manuscripts with independent citation requests (n = 102; 27.3% of manuscripts). A significant interaction was observed between the presence of self-citation requests and the likelihood of any citation request being incorporated (100% incorporation in manuscripts with self-citation requests alone versus 62.7–72.2% with any independent citation request; Fisher's exact test p<0.0005).

**Data Availability Statement:** All relevant data are contained in the Supporting Information file.

**Funding:** EP, MS and BH are employees of Kainic Medical Communications Ltd. (www.kainicmedical.com) and were collectively responsible for the

study design, data collection and analysis, decision to publish and preparation of this manuscript. BH is a director and owner of Kainic Medical Communications Ltd. The publication fee for this manuscript was funded by Kainic Medical Communications Limited.

**Competing interests:** We have read the journal's policy and the authors of this manuscript have the following competing interests: EP, MS and BH are employees of Kainic Medical Communications Ltd, a medical communications agency that provides medical writing support and consultancy services to authors submitting manuscripts to peer reviewed journals. BH is a director and owner of Kainic Medical Communications Ltd. This does not alter our adherence to PLOS ONE policies on sharing data and materials.

## Conclusions

Requests for self-citations during the peer review process are common. The transparency of open peer review may have the unexpected effect of encouraging authors to incorporate self-citation requests by disclosing peer reviewer identity.

## Introduction

Peer reviewers are invited to aid journal editors in determining whether a manuscript merits publication by providing suggestions to authors on producing sound research, improving the quality of their publications and providing specific expertise in a subject [1, 2]. However, the peer review process is not without flaws [1, 3]. These flaws, and potential methods of mitigating their impact, have been widely debated, but have rarely been objectively assessed [3]. For example, while peer review is intended to be a sober and objective process, subjective judgement and opinion cannot be avoided [3].

Peer review is a volunteer effort, but also offers opportunities for reviewers to seek personal reward and influence the direction and content of others' publications. In particular, peer reviewers may benefit from including requests for their own work to be cited in peer review reports as it increases their citation counts [1, 4–7]. Self-citations have been reported by approximately one quarter of authors of scientific journal articles and are estimated to account for 12% of all requests for additional citations made during peer review [5, 8].

Given that peer reviewers are experts in their field, self-citation requests are an expected, justified and necessary part of peer review [4]. However, the Committee on Publication Ethics (COPE) Ethical Guidelines for Peer Reviewers state that reviewers should not suggest citations to their own work unless there is a valid reason and the citation is required to fill in gaps or enhance the quality of a manuscript [2]. Therefore, authors are expected to judge the suitability of any request for additional citations and amend the manuscript or rebut the request accordingly, but many authors instead consider it expeditious to incorporate citations requested by peer reviewers, rather than debate their merit [1, 3, 9]. In fact, 70% of self-citation requests were found to be incorporated into published manuscripts during blinded peer review, with 25% of published manuscripts ultimately including a self-citation suggested by a reviewer [5].

Open peer review allows authors and the readership to know the identity of peer reviewers [3]. Although much of the focus on the impact of open peer review has been on peer reviewer behaviour [3, 10], a process that makes authors' responses and amendments available for public scrutiny may encourage authors to justify their amendments or provide rebuttal for a broad audience of potential readers, rather than merely incorporating amendments to satisfy peer reviewers/journal editors alone. Accordingly, an open peer review process may place greater onus on authors to consider the purpose and value of additional citations suggested during peer review. Notably, the impact of open peer review on peer reviewer and author biases has been considered to be a 'high priority' area for research into peer review [3].

This study aimed to investigate peer reviewer and author behaviour regarding requests for self-citation during peer review by assessing the prevalence of, and accession to, requests for self-citation in a general medicine journal applying an open peer review process.

## Methods

Publicly accessible peer review reports from *BMC Medicine*, a high-impact general medicine journal, were retrospectively reviewed for manuscripts published between 1 January 2017 and 31 December 2018. *BMC Medicine* was selected because it has an open peer review process and a wide scope, offering a reduced risk of therapy area, geographical and other biases compared with a more specialised journal [3]. All manuscripts published during the study timeframe were included in this study except for manuscripts that had not undergone peer review, had been retracted or had one or more inaccessible reviews (e.g. supplementary file uploads, which were not readily accessible online, or reports had not been uploaded). All reviewer reports published during this timeframe were performed using open peer review methodology, wherein the identity of peer reviewers was disclosed to the authors in all peer review reports. Peer review reports, and the identity of the reviewer responsible for each report, were subsequently published online alongside accepted manuscripts.

Data collected included: article type; gender of the reviewer; geographical region the reviewer resided in; number of citations requested by the reviewer; number of self-citations; and if the reviewer disclosed an interest in any self-citations (see S1 File). All references mentioned in the peer review reports were considered requests for citation, except for those cited in the originally submitted manuscript.

Self-citations were defined as reference requests that were authored or co-authored by the reviewer. Reference requests that were insufficiently detailed to allow reference identification were presumed to not involve a self-citation. A self-citation was considered disclosed if the language used in the report clearly identified a personal interest of the reviewer in the citation, for example, using language such as "our study," "we," "I," "my," etc. If the reasonable author would not be expected to know that the reviewer had an interest in the reference requested under blinded conditions (for example, the reviewer's name being visible in the requested citation but the reviewer did not explicitly state their interest) the request was recorded as undisclosed.

All rounds of peer review for each manuscript were examined. Multiple rounds of peer review performed by one reviewer were considered to represent one report. The primary endpoint was the proportion of self-citations incorporated into published manuscripts compared with independent citations. Secondary endpoints included the proportion of all peer reviews with requests for self-citation versus requests for independent citations; the relative proportion of manuscripts with requests for self-citation versus independent citations incorporated into the published manuscript; and the proportion of requests for self-citation that were disclosed.

A chi-square test was performed to compare the proportion of self-citations versus independent citations incorporated into published manuscripts. The test was expected to have a statistical power of at least 0.85 to detect a 15% difference between the proportion of self-citation and independent reference requests being incorporated into manuscripts with $\alpha = 0.05$ based on the following assumptions: the total sample size would be approximately 900 peer review reports; the number of references requested be approximately two-thirds of the total number of peer reviews (n = 600) [5]; the ratio of independent references versus self-citations is 7:3 [5]; and 70% of self-citations would be incorporated [5] versus a hypothesised 85% of independent reference requests being incorporated given that these requests could be predicted to be subject to a lower level of selection bias, and therefore of higher relevance and greater likelihood of incorporation. The proportion of published manuscripts that incorporated $\geq 1$ citation request was compared using the Fisher's exact test using the Freeman-Halton extension because of the low number in one of the categories in the contingency table.

## Results

Overall, 466 published manuscripts were reviewed, of which 373 met the study eligibility criteria. In total, 932 peer review reports were included in this analysis (Fig 1). Males comprised approximately two thirds of reviewers. Nearly half of reviewers were based in Europe and one third in the United States of America. Most articles were original research (Table 1).

Requests for a total of 581 citations to be incorporated were made across 241 (25.9%) reports. At least one self-citation request was present in 107 reports (44.4% of reports with a citation request; 11.5% of all reports), comprising 33.6% (n = 195) of all requested citations. Three hundred and eighty-six (66.4%) independent citations were requested. Most reports with requests for self-citation comprised a single self-citation request (median, 1; range 1–8). 65.1% (n = 127) of self-citation requests were incorporated into the final published manuscript. Requests for self-citation were significantly more likely to be incorporated in the final manuscript than independent citation requests (52.1% [n = 201]; chi-square p = 0.003) (Fig 2). The reviewer's involvement in self-citations was disclosed in only 15 (14.0%) instances (Fig 1).

At the manuscript level, self-citation requests were also less prevalent than requests for independent citation during peer review (n [%] of manuscripts, 94 [25.2] versus 149 [39.9]). However, self-citation requests were incorporated in 72 published manuscripts (76.6% of manuscripts with requests; 19.3% of all manuscripts) versus 102 manuscripts with requests for independent citations (68.5% of manuscripts with requests; 27.3% of all manuscripts; Fig 1).

A statistically significant association between citation request classification (self-citation only, independent citation only or both; p<0.0005 using Fisher's exact test) and inclusion was observed at the manuscript level (Fig 3). Of the 35 manuscripts where self-citation requests alone were made, all manuscripts had a peer reviewer-requested citation incorporated into the published version compared with 72.2% (n = 65/90) of manuscripts with independent citation requests alone and 62.7% (n = 37/59) of manuscripts with both self-citation and independent citations requested during review.

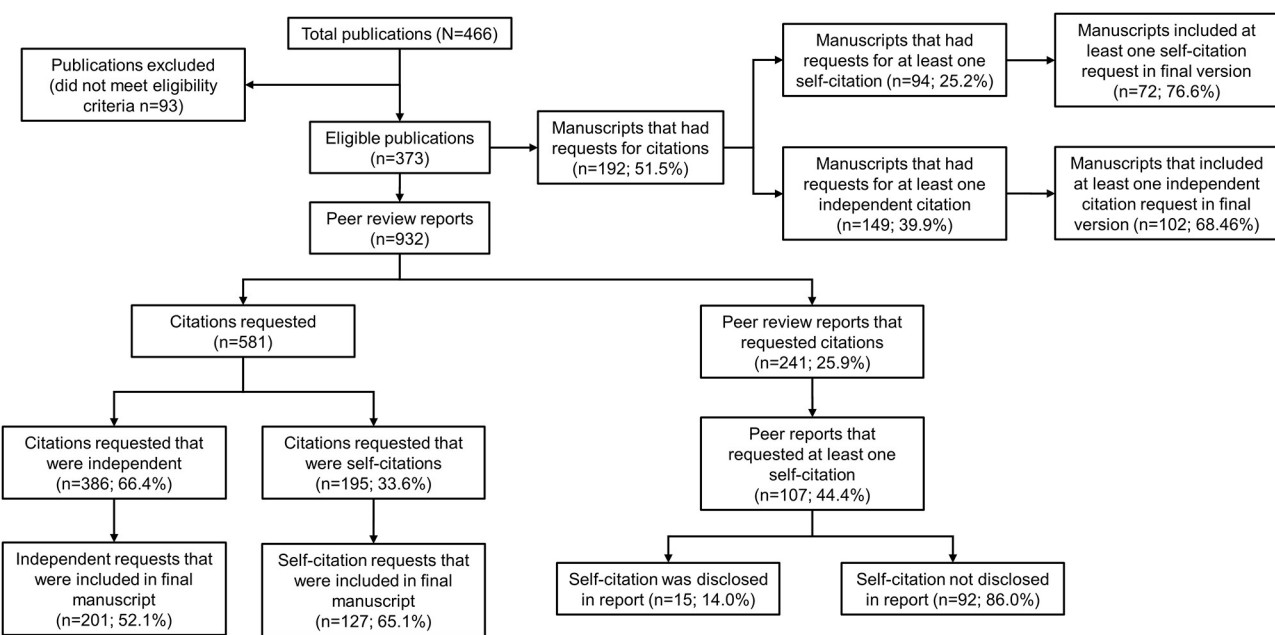

**Fig 1. Flow diagram of citation requests, reviewers requesting self-citations and accession in the final published manuscript.**

**Table 1. Peer reviewer and manuscript characteristics.**

|  | n | % |
| --- | --- | --- |
| **Total number of publications** | 373 | - |
| **Total number of peer review reports** | 932 | - |
| **Gender** |  |  |
| Male | 613 | 65.8 |
| Female | 305 | 32.7 |
| Unknown | 14 | 1.5 |
| **Region** |  |  |
| USA | 255 | 27.4 |
| Europe | 434 | 46.6 |
| North America (excl. USA) | 50 | 5.4 |
| South America | 22 | 2.4 |
| Asia | 40 | 4.3 |
| Middle East | 10 | 1.1 |
| Africa | 16 | 1.7 |
| Oceania | 66 | 7.1 |
| Multiple regions | 19 | 2.0 |
| Unknown | 20 | 2.1 |
| **Article type** |  |  |
| Commentary | 3 | 0.8 |
| Correspondence | 9 | 2.4 |
| Debate | 9 | 2.4 |
| Guideline | 3 | 0.8 |
| Minireview | 12 | 3.2 |
| Opinion | 24 | 6.4 |
| Original research | 299 | 80.2 |
| Review | 13 | 3.5 |
| Technical advance | 1 | 0.3 |

## Discussion

This retrospective study of open peer review reports in a general medicine journal indicated that approximately one third of requests for additional citations to be incorporated into a manuscript during peer review were for the reviewer's own work to be cited. The reviewer's interest in these requests generally remained undisclosed, and ultimately, one in five manuscripts incorporated at least one peer reviewer self-citation in the final publication.

Approximately two thirds of self-citation requests were incorporated in the final published manuscript, which is significantly greater than the approximately half of all independent citation requests incorporated. At the manuscript level, any request for self-citation during peer review was also significantly more likely to be acceded to than any request for an independent citation.

Applying Ockham's razor suggests an unspoken quid pro quo during the peer review process; that is, any request for self-citation made by a peer reviewer offering a positive recommendation should be incorporated into the final manuscript. This may be particularly evident when the identity of the reviewer is disclosed to the author(s) during peer review and can be easily linked with the authorship of suggested citations. Notably, positive peer review reports (recommending revision and resubmission or acceptance) have previously been found to be twice as likely than negative reports (recommending rejection) to contain self-citation requests

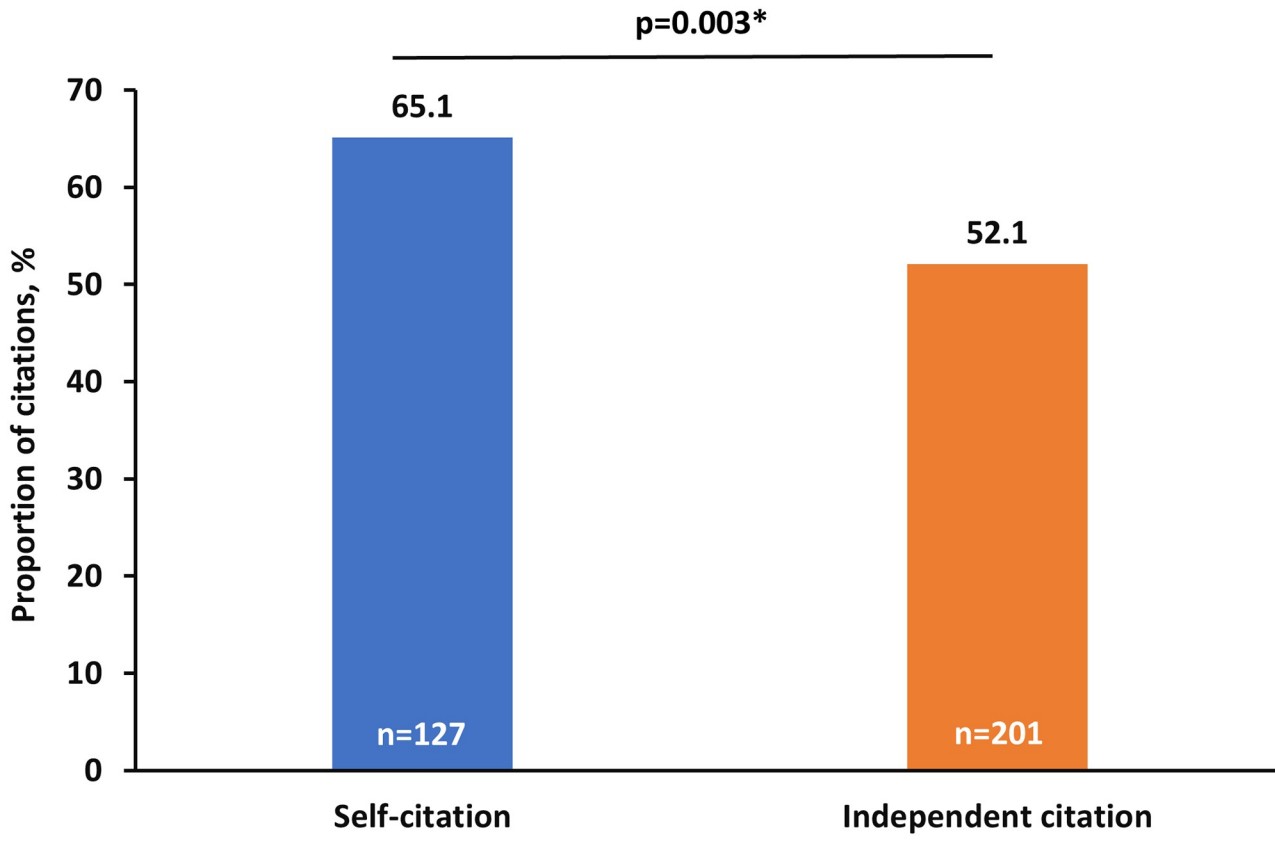

**Fig 2. Comparison of individual independent and self-citations raised in peer review being incorporated in published manuscripts.** *Chi-square test.

[5, 10]. However, this may be a reflection of rejected manuscripts containing major flaws in study design, hence the reason for rejection, and it follows the narrative sections may have received fewer suggestions for improvement, including citation suggestions [5]. Likewise, in this study self-citations were included without exception when presented in isolation, suggesting a lack of critical review for relevance by authors. By comparison, only 62.7–72.2% of manuscripts with any independent citation requested during peer review incorporated any one of the requested citations. In fact, the presence of an independent citation request appears to have spurred critical analysis of self-citations given the lower accession rates for manuscripts with requests for both self- and independent citations to be inserted.

Our base findings are consistent with an earlier study of blinded peer review that reported 44% of requests for additional citations during peer review included at least one peer reviewer self-citation (12% of all reports) [5] and a study of open peer review reporting self-citation in 13.3% of reports [5, 10]. Likewise, the overall proportion of requests for additional citations comprising self-citations was comparable (34% versus 29–32%) [5, 10]. This confirmed the previous report of open versus blinded peer review not appearing to influence peer reviewer behaviour in this regard [10]. This is contrary to prior assertions that open peer review may disincentivise reviewers unnecessarily requesting their own papers for citation and give authors autonomy over deciding whether accession to a self-citation request is justified [1, 3, 5, 7], but is consistent with open peer review not adversely affecting, nor improving, the quality of peer review [9, 10].

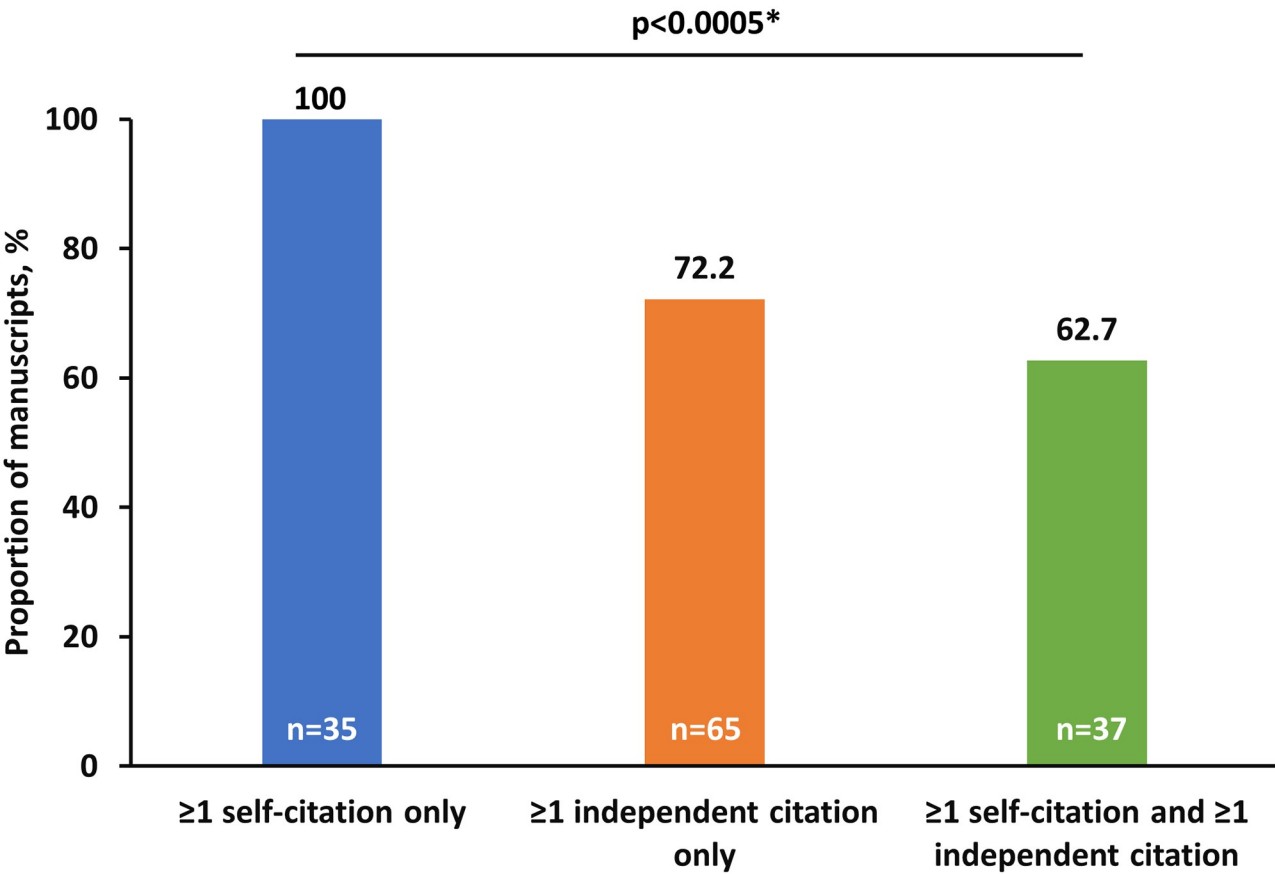

**Fig 3. Manuscript-level comparison of citations raised during peer review being incorporated in the published manuscripts.** *Fisher's exact test using the Freeman-Halton extension.

While the results of this study and that of Thombs and colleagues (2015) are consistent, key differences in study design and reporting should be noted [5, 10]. In particular, the earlier study should be interpreted with caution given that the manuscript was authored by editorial staff of the journal studied, who were likely involved in both peer reviewer selection and editorial decisions surrounding manuscript publication [5, 10]. The manuscripts were also subsequently published in the same journal as was investigated and the scope was limited to psychosomatic conditions, a much narrower field with a limited pool of peer reviewers compared with the manuscripts examined here [5, 10].

Furthermore, no comparison was made between the incorporation of self- versus independent citations in the earlier study to provide context [5, 10]. Although the current study only involved manuscripts that were subsequently accepted for publication, whereas the earlier study included manuscripts that were both accepted and rejected [5], it is possible that declining to incorporate requested amendments could increase the risk of rejection, resulting in an overestimation of author incorporation rates in this study.

Mandatory disclosure of interests in suggested citations has been proposed as a method of influencing the prevalence and incorporation of self-citation requests during peer review [1, 7] because only a small proportion of peer reviewers voluntarily disclose self-citations. In addition, a brief rationale of why the self-citation requested is relevant and important to the integrity of the manuscript should be provided, but the merits of inclusion should be up for debate

without prejudicing the chance of the manuscript being accepted for publication [1, 7]. One suggestion has been that editors may wish to communicate to authors which peer reviewer comments are considered critical to address compared with those of lesser importance [3], although the feasibility of this approach, given the burden it would place on individual editors, is questionable. However, it is possible disclosure of a peer reviewer's involvement in a citation could have an unintended effect of increasing self-citation by highlighting such instances to authors who see agreeing with such requests as necessary to achieve publication, as demonstrated by all authors acceding to peer reviewer requests for self-citation when they were made in isolation here [1, 4, 6].

Notably, in May 2020, the publisher of *BMC Medicine* altered their peer review methodology, opting for "transparent" versus "open" peer review, indicating that the names of reviewers will no longer be disclosed to the authors or general public [11]. Instead, anonymised peer review reports are to be provided to authors and published online [11]. The stated rationale for this change focused on peer reviewer anonymity potentially improving peer reviewer recruitment [11]. No indication of revealing peer reviewer identity to authors during the peer review process potentially affecting the integrity of the peer review process was raised [11].

This study has several limitations, including the sample being derived from a single journal and limited to accepted publications [3]. Peer review reports may also have been vetted by the editors prior to dissemination, so some requests for citations included in the original versions of reports may not be present in the publicly available versions [3, 5]. Other reports were unable to be accessed. Furthermore, while assessing how experience may affect peer reviewer behaviour was considered to be desirable, deriving a fair measure of experience using factors such as number of publications, prior peer reviews, academic rank, length of time in the field and the extent of expertise in the relevant field was not considered feasible [3]. In addition, this study did not investigate the validity of any rationale or justification for inclusion of additional references due to inability of individual investigators to provide subjective judgement on the relevance of requested citations across the range of subject areas covered in a general medical journal and the expectation that academic discourse between authors and peer reviewers should, in theory, arbitrate the relevance of any requested citations.

## Conclusions

Requests for self-citation are prevalent in positive peer review reports. Contrary to previous conjecture, open peer review does not appear to decrease the prevalence of self-citation requests by peer reviewers. Instead, by disclosing the identity of the peer reviewer to authors during open peer review, the process may unwittingly undermine academic debate as authors consider acceding to self-citation requests to be the most expeditious route to achieving publication.

## Supporting information

**S1 File. Peer reviewer self-citation analysis raw data.**
(XLSX)

## Acknowledgments

We would like to acknowledge Dr Nic Rawlence from the Department of Zoology, University of Otago (Dunedin, New Zealand) for reviewing the draft manuscript and providing editorial comments.

## Author Contributions

**Conceptualization:** Erin Peebles, Marissa Scandlyn, Blair R. Hesp.

**Data curation:** Erin Peebles, Marissa Scandlyn.

**Formal analysis:** Erin Peebles.

**Funding acquisition:** Blair R. Hesp.

**Investigation:** Erin Peebles, Marissa Scandlyn.

**Methodology:** Erin Peebles, Marissa Scandlyn, Blair R. Hesp.

**Project administration:** Blair R. Hesp.

**Resources:** Blair R. Hesp.

**Supervision:** Blair R. Hesp.

**Validation:** Erin Peebles, Marissa Scandlyn.

**Visualization:** Erin Peebles, Blair R. Hesp.

**Writing – original draft:** Erin Peebles.

**Writing – review & editing:** Erin Peebles, Marissa Scandlyn, Blair R. Hesp.

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
