## [Decision Letter · Decision Letter 0]

7 Jul 2020

PONE-D-20-13025

A retrospective study investigating requests for self-citation during open peer review in a general medicine journal

PLOS ONE

Dear Dr. Hesp,

Thank you for submitting your manuscript to PLOS ONE. After careful consideration, we feel that it has merit but does not fully meet PLOS ONE’s publication criteria as it currently stands. Therefore, we invite you to submit a revised version of the manuscript that addresses the points raised during the review process.

Your work has been refereed by two acknowledged reviewers in the field of scientific publishing. Overall, they raised good comments on your manuscript, but also expressed some key concerns, especially related to specific features of the self-citation requests and the discussion of the study outcomes, that you must address during your revisions of the paper.

We look forward to receiving your revised manuscript.

Kind regards,

Sergio A. Useche, Ph.D.

Academic Editor

PLOS ONE

Journal Requirements:

2. Please clarify whether the reviewers' name is known to the authors when they receive a first decision or only at the end of the review process

"We have read the journal's policy and the

authors of this manuscript have the following competing interests: EP, MS and BH are

employees of Kainic Medical Communications Ltd, a medical communications agency

that provides medical writing support and consultancy services to authors submitting

manuscripts to peer reviewed journals. BH is a director and owner of Kainic Medical

Communications Ltd."

Reviewers' comments:

Reviewer's Responses to Questions

**Comments to the Author**

1. Is the manuscript technically sound, and do the data support the conclusions?

Reviewer #1: Partly

Reviewer #2: Partly

2. Has the statistical analysis been performed appropriately and rigorously? 

Reviewer #1: N/A

Reviewer #2: I Don't Know

3. Have the authors made all data underlying the findings in their manuscript fully available?

Reviewer #1: Yes

Reviewer #2: Yes

4. Is the manuscript presented in an intelligible fashion and written in standard English?

Reviewer #1: Yes

Reviewer #2: Yes

5. Review Comments to the Author

Reviewer #1: Although the topic is interesting but there is some concerns that authors should consider,

1- The reviewers asked just for citing their papers or a mixture of papers in the literature including their work?

2- Are the suggested papers related to manuscript or no?

Reviewer #2: The manuscript “A retrospective study investigating requests for self-citation during open peer review in a general medicine journal” by E. Peebles, M. Scandlyn, and B.R. Hesp, studied the prevalence of reviewers’ self-citation requests and authors’ behavior regarding their incorporation into published manuscripts in BMC Medicine, a journal with open peer review. So far, a few studies analyzed the rate of peer review requests for citation of their own work, so the submitted manuscript provides novel data on the topic that certainly needs more investigation.

The manuscript is generally well presented, but there are several issues which needs to be resolved to improve the quality of data presentation and interpretation.

Methods:

1. Details on the peer review process in the BMC Medicine should be briefly outlined. There are variations of open peer review (open vs. transparent ). BMC Medicine declares a transparent peer-review system, which does not include the disclosure of the names of the reviewers (https://bmcmedicine.biomedcentral.com/submission-guidelines/peer-review-policy). So, it is a bit confusing, because identities of the reviewers are available alongside the published articles. It would be worth mentioning whether reviewer identities were available for all assessed articles, and at which stage their identities were disclosed to authors. This relates to the rationale for dividing the self-citation requests to disclosed and undisclosed regarding the reviewers’s statements. If the names of the reviewers are uniformly publicly disclosed, being aware that their requests for citations will also be publicly available, reviewers potentially do not find disclosure of interest necessary.

Results:

1. Figure 1: In the flow-chart, the proportions are calculated for numbers in the previous levels of the chart. It would be useful to include the proportions among the total number of items as well.

2. Figure 2: Reading of the main text was required for the interpretation, so legend requires more detailed specification of presented data to improve the clarity.

3. Figure 2B, Fisher’s exact test uses a 2x2 contingency table, whereas larger tables are usually assessed by Chi-square test. It should be specified which differences (and how) were tested in the figure legend.

4. Page 8 ln 146, A word “interaction” is mostly used to interpret the results of a two-way ANOVA, and describe the potential combined and dependent effects of different factors on a certain outcome. Suggested alternatives are: significant relationship between classification factors, or significant difference in proportions.

Discussion

1. Pg 8 ln 157: Authors state, “The reviewer’s interest in these requests generally remained undisclosed…”. This statement is correct, but the fact that reviewer’s identity is eventually disclosed poses the question of relevance of omitting such disclosure statement.

2. Disclosure of interest is further discussed on the page 10 ln, 198, but the data based on a single parameter should be interpreted with caution. Proper assessment of the reviewers intent should be also based on the rationale behind the reviewer’s suggestion, which was not assessed in this study. The peer review aims to improve the quality of published articles, and under this assumption, the reviewer will honestly and objectively suggest the inclusion of omitted studies which he/she considers relevant for the topic, and provide a valid rationale for such request. Assessment of reviewers’ rationales would strengthen the conclusions on their intents, as well as authors' responses.

3. Pg 9, ln 169: It should be taken into account that the higher proportions of self-citation requests in positive peer reviews might reflect the fact that rejected manuscripts are burdened with major flaws in study design and methods, which were the reason for rejection, so their narrative parts (i.e. Discussion) received less suggestions for further improvement (including omissions of relevant studies).

4. Authors compare their findings with the study by Thombs BD et al. (reference 5), which is thoroughly discussed. The same group published a study of self-citation by peer reviewers in a journal with single-blind peer review vs. journal with open peer review (Levis AW et al. J Psychosom Res. 2015;79:561-5), so it is not clear why it was omitted, because it even better relates to the design of this study.

Minor point: Figure legends should be placed at the end of the article.

6. PLOS authors have the option to publish the peer review history of their article (what does this mean?). If published, this will include your full peer review and any attached files.

Reviewer #1: No

Reviewer #2: **Yes: **Natasa Kovacic

---

## [Author Response · Author response to Decision Letter 0]

8 Jul 2020

Dear Dr Useche,

Thank you for the feedback provided on this manuscript and requesting that we submit a revised manuscript for consideration for publication in PLOS ONE.

We have reviewed the feedback provided by the Reviewers and have revised the manuscript accordingly, as detailed in the marked-up copy of the manuscript submitted alongside this response.

We have addressed each of the comments specifically as detailed below. Please note that all page and line numbers are for the version of the manuscript with tracked changes.

Yours sincerely,

Blair Hesp

Comment #1:

Please ensure that your manuscript meets PLOS ONE's style requirements, including those for file naming. The PLOS ONE style templates can be found at:

Response #1:

We have reviewed PLOS ONE’s style requirements and applied the necessary amendments throughout the manuscript.

Comment #2:

Please clarify whether the reviewers' name is known to the authors when they receive a first decision or only at the end of the review process

Response #2:

 The following statement has been inserted on page 5, lines 85–88, to increase clarity:

“All reviewer reports published during this timeframe were conducted using open peer review methodology, wherein the identity of peer reviewers was disclosed to the authors in all peer review reports. Peer review reports, and the identity of the reviewer responsible for each report, were subsequently published online alongside accepted manuscripts.”

Further details that are highly relevant to this comment are outlined in response to Comment #2.1 by Reviewer #2. 

Comment #3:

Thank you for stating the following in the Competing Interests section:

"We have read the journal's policy and the authors of this manuscript have the following competing interests: EP, MS and BH are employees of Kainic Medical Communications Ltd, a medical communications agency that provides medical writing support and consultancy services to authors submitting manuscripts to peer reviewed journals. BH is a director and owner of Kainic Medical Communications Ltd."

Response #3:

We can confirm that none of the authors’ competing interests alter our adherence to all PLOS ONE policies on sharing data and materials.

The required declarations have been incorporated in an updated version of our original cover letter that has been uploaded via the Editorial Manager submission system.

Comment #4:

Please include captions for your Supporting Information files at the end of your manuscript, and update any in-text citations to match accordingly.

Response #4:

A link to the supporting information has now been inserted on page X, lines and a corresponding in-text reference inserted on page 5, line 91.

Reviewer #1:

Comment #1.1:

The reviewers asked just for citing their papers or a mixture of papers in the literature including their work?

Response #1.1:

We agree with the reviewer’s assessment of the need to provide a comparison of manuscripts where independent citations, self-citations and a mixture were requested. Accordingly, we believe that the reviewer’s comment is addressed by Figure 1, which presents such data and no further amendment is necessary to address this comment. 

Comment #1.2:

Are the suggested papers related to manuscript or no?

Response #1.2:

The authors agree this is an important point, but given that a key premise of this study was to observe and report author behaviour when those authors had full knowledge of the peer reviewers’ identities, we do not believe that subjective assessments of the validity of any citation request made by a peer reviewer would be appropriate in this study. 

Firstly, any subjective assessment of the validity of a reference is best made by the authors who are likely experts in their field and are best placed to judge the relevance of a citation to their study. Secondly, we believe that the investigators in the current study must remain neutral observers making objective assessments.

However, the following amendments have been made on page 12, lines 236–241, to provide greater clarity and justification for this approach:

“In addition, this study did not investigate the relevance validity of any rationale or justification for inclusion of additional references due to inability of individual investigators to provide subjective judgement on the relevance of requested citations across the range of subject areas covered in a general medical journal and the expectation that academic discourse between authors and peer reviewers should, in theory, arbitrate the relevance of any requested citations,. instead relying on the authors’ judgement.”

Reviewer #2

METHODS

Comment #2.1:

Details on the peer review process in the BMC Medicine should be briefly outlined. There are variations of open peer review (open vs. transparent ). BMC Medicine declares a transparent peer-review system, which does not include the disclosure of the names of the reviewers (https://bmcmedicine.biomedcentral.com/submission-guidelines/peer-review-policy). So, it is a bit confusing, because identities of the reviewers are available alongside the published articles. It would be worth mentioning whether reviewer identities were available for all assessed articles, and at which stage their identities were disclosed to authors. This relates to the rationale for dividing the self-citation requests to disclosed and undisclosed regarding the reviewers’s statements. If the names of the reviewers are uniformly publicly disclosed, being aware that their requests for citations will also be publicly available, reviewers potentially do not find disclosure of interest necessary.

Response #2.1:

The authors thank the reviewer for raising this important point. BMC Medicine changed their peer-review methodology on or after 4 May 2020 from an “open” to “transparent” peer review model (see: https://blogs.biomedcentral.com/bmcblog/2020/05/04/transparency-openness-and-peer-review/).

The articles assessed in this study were published between 1 January 2017 and 31 December 2018, when “open” peer review methodology was used. The publisher has defined “open” peer review as a process wherein the names of peer reviewers are included in all reports and published alongside accepted articles, whereas “transparent” peer review sees anonymised reports provided to the authors and published online.

In addition, all the reviewer reports during this timeframe investigated in this study included the statement “I understand that my name will be included on my report to the authors and, if the manuscript is accepted for publication, my named report including any attachments I upload will be posted on the website along with the authors' responses”, confirming that open peer review methodology was used during this time.

Therefore, in light of this change in policy, we have added the following statement on page 5, lines 85–88:

“All reviewer reports published during this timeframe were conducted using open peer review methodology, wherein the identity of peer reviewers was disclosed to the authors in all peer review reports. Peer review reports, and the identity of the reviewer responsible for each report, were subsequently published online alongside accepted manuscripts.”

Furthermore, we have added the following additional statement in the Discussion outlining this recent change, the publisher’s rationale and how this relates to the outcomes reported in this study on page 11, lines 222–228:

“Notably, in May 2020, the publisher of BMC Medicine altered their peer review methodology, opting for “transparent” versus “open” peer review, indicating that the names of reviewers will no longer be disclosed to the authors or general public. [11] Instead, anonymised peer review reports are to be provided to authors and published online. [11] The stated rationale for this change focused on peer reviewer anonymity potentially improving peer reviewer recruitment. [11] No indication of revealing peer reviewer identity to authors during the peer review process potentially affecting the integrity of the peer review process was raised. [11]”

New reference 11 has also been added, which directs the reader to the publisher’s blog post announcing the change in their peer review policy.

RESULTS

Comment #2.2:

Figure 1: In the flow-chart, the proportions are calculated for numbers in the previous levels of the chart. It would be useful to include the proportions among the total number of items as well.

Response #2.2:

Appropriately addressing this comment is difficult given the complexity of Figure 1. The current presentation was adopted in an effort to maintain simplicity and a natural interpretation of the data. We also believe that providing absolute percentages based on the overall study population adds negligible incremental value to the interpretation of the study and risks confusing the reader. Namely, describing the overall percentage of manuscripts with an undisclosed self-citation is a limited value whereas illustrating the percentage of self-citations that are not disclosed is of high interest. This is also not possible when referring to citations versus manuscripts. Likewise, it is not possible maintain an accurate description when presenting both sets of data because the overall population would require describing manuscripts with a self-citation and that self-citation being incorporated versus the percentage of self-citations incorporated.

Furthermore, where necessary and relevant, effort is already given to present percentages based on the overall population and as a proportion of the relevant subgroup. For example, the percentage of all citation requests that were self-citations and the percentage of all manuscripts with a self-citation are presented on page 7, lines 132–134 and page 8, lines 148–1451.

Comment #2.3:

Figure 2: Reading of the main text was required for the interpretation, so legend requires more detailed specification of presented data to improve the clarity.

Response #2.3:

Given the 15-word limit for figure titles, we have now split Figure 2 into Figures 2 and 3 and amended the figure titles accordingly.

The title for Figure 2 on page 7, lines 141–144 now reads:

“Comparison of individual independent and self-citations raised in peer review being incorporated in published manuscripts.”

The title for Figure 3 on page 7, lines 158–159 now reads:

“Manuscript-level comparison of citations raised during peer review being incorporated in published manuscripts.”

Mention of Figures 2A and 2B have now been replaced with Figures 2 and 3, respectively on page 8, line 139 and 154.

Comment #2.4:

Figure 2B, Fisher’s exact test uses a 2x2 contingency table, whereas larger tables are usually assessed by Chi-square test. It should be specified which differences (and how) were tested in the figure legend.

Response #2.4:

A Fisher’s exact test can be used in larger contingency tables using the Freeman-Halton extension. Fisher’s exact test was applied to the data presented in Figure 2B because a value in one of the columns = 0, so a Chi-square test is unable to be performed.

We have amended the figure legend on page 8, line 160 as follows to clarify this:

“**Fisher’s exact test using the Freeman-Halton extension”

The Methods have also been amended on page 6, lines 118–121 to read:

“A Fisher’s exact test was used to compare tThe proportion of published manuscripts that incorporated ≥1 citation request, was compared using the Fisher’s exact test using the Freeman-Halton extension because of due to the low number in one of the categories in the contingency table.”

Comment #2.5:

Page 8 ln 146, A word “interaction” is mostly used to interpret the results of a two-way ANOVA, and describe the potential combined and dependent effects of different factors on a certain outcome. Suggested alternatives are: significant relationship between classification factors, or significant difference in proportions.

Response #2.5:

The word “interaction” was chosen in this instance after consulting the literature on how to best express the results of a statistical test that compared three categories, as opposed to making a head-to-head comparison. However, we understand the reviewer’s concerns about potential confusion surrounding the use of this term in this context and have amended the text on page 8, lines 152–154, as follows:

“A statistically significant association interaction between citation requests classification (self-citation only, independent citation only or both; p<0.0005 using Fisher’s exact test) and the incorporation of citations was observed at the manuscript level (Fig 2B).”

DISCUSSION

Comment #2.6:

Pg 8 ln 157: Authors state, “The reviewer’s interest in these requests generally remained undisclosed…”. This statement is correct, but the fact that reviewer’s identity is eventually disclosed poses the question of relevance of omitting such disclosure statement.

Response #2.6:

We defined disclosure (page 5, lines 96–101) as circumstances under which we believe a casual reader, whether they be a naive author, a journal editor or third party reviewing an open peer review report, could not be reasonably expected to know that the reviewer had an interest in the requested citation, despite being aware of the reviewers identity at the time of reading.

We believe this is important because the reasonable author will be expected to retrieve and consider any citation requested by a peer reviewer. This involves a degree of effort beyond reading the peer review report, and it is only after applying this additional effort that we believe the link between the reviewer and citation could reasonably be expected to be drawn. Therefore, we believe that the criteria we have used to define non-disclosure are appropriate and amending the manuscript by removing this content would ultimately remove important information from the body of scientific knowledge.

Comment #2.7:

Disclosure of interest is further discussed on the page 10 ln, 198, but the data based on a single parameter should be interpreted with caution. Proper assessment of the reviewers intent should be also based on the rationale behind the reviewer’s suggestion, which was not assessed in this study. The peer review aims to improve the quality of published articles, and under this assumption, the reviewer will honestly and objectively suggest the inclusion of omitted studies which he/she considers relevant for the topic, and provide a valid rationale for such request. Assessment of reviewers’ rationales would strengthen the conclusions on their intents, as well as authors' responses.

Response #2.7:

With reference to our response to Comment #1.2 from Reviewer #1, we believe that it would be inappropriate to make subjective judgements about whether a requested citation is appropriate. The authors are likely to be in the best position to make this judgement and, as investigators, we must report objective outcomes based on statistical analysis without introducing a risk of bias resulting from each investigator’s subjective opinion of relevance based on differing background knowledge and experience. The overall feasibility of assessing the relevance of citations in this study is also highlighted by the requirement for non-experts to assess the relevance of 581 citations relating to 241 manuscripts in a general medical journal.

Therefore, we have made the following amendments on page 12, lines 236–241, to provide greater clarity and justification for this approach:

“In addition, this study did not investigate the relevance validity of any rationale or justification for inclusion of additional references due to inability of individual investigators to provide subjective judgement on the relevance of requested citations across the range of subject areas covered in a general medical journal and the expectation that academic discourse between authors and peer reviewers should, in theory, arbitrate the relevance of any requested citations,. instead relying on the authors’ judgement.”

Comment #2.8:

Pg 9, ln 169: It should be taken into account that the higher proportions of self-citation requests in positive peer reviews might reflect the fact that rejected manuscripts are burdened with major flaws in study design and methods, which were the reason for rejection, so their narrative parts (i.e. Discussion) received less suggestions for further improvement (including omissions of relevant studies)

Response #2.8:

We agree with the comments from the reviewer, and as a result have added the following text on page 9, lines 177–180:

“However, this may be a reflection of rejected manuscripts containing major flaws in study design, hence the reason for rejection, and it follows the narrative sections may have received fewer suggestions for improvement, including citation suggestions.”

Comment #2.9

Authors compare their findings with the study by Thombs BD et al. (reference 5), which is thoroughly discussed. The same group published a study of self-citation by peer reviewers in a journal with single-blind peer review vs. journal with open peer review (Levis AW et al. J Psychosom Res. 2015;79:561-5), so it is not clear why it was omitted, because it even better relates to the design of this study.

Response #2.9

Thank you for bringing this manuscript to our attention. This is indeed a relevant citation that should be discussed in the context of this study and has been added to the manuscript as reference 10, and cited, where relevant.

In particular, the following additional text has been added to the Discussion on page 10, lines 186–195:

“Our base findings are consistent with an earlier study of blinded peer review that reported 44% of requests for additional citations during peer review included at least one peer reviewer self-citation (12% of all reports) [5] and a study of open peer review reporting self-citation in 13.3% of reports. [5,10] Likewise, the overall proportion of requests for additional citations comprising self-citations was comparable (34% versus 29–32%). [5,10] Notably,This confirmed the previous report of open versus blinded peer review did not appearing to influence peer reviewer behaviour in this regard.[10]“

Minor point:

Minor point: Figure legends should be placed at the end of the article.

Response:

Please note that we have inserted the figure captions in the main body of the text in line with the instructions provided by PLOS ONE here: https://journals.plos.org/plosone/s/figures.

---

## [Decision Letter · Decision Letter 1]

4 Aug 2020

A retrospective study investigating requests for self-citation during open peer review in a general medicine journal

PONE-D-20-13025R1

Dear Dr. Hesp,

We’re pleased to inform you that your manuscript has been judged scientifically suitable for publication and will be formally accepted for publication once it meets all outstanding technical requirements.

Kind regards,

Sergio A. Useche, Ph.D.

Academic Editor

PLOS ONE

Additional Editor Comments (optional):

Reviewers' comments:

Reviewer's Responses to Questions

**Comments to the Author**

1. If the authors have adequately addressed your comments raised in a previous round of review and you feel that this manuscript is now acceptable for publication, you may indicate that here to bypass the “Comments to the Author” section, enter your conflict of interest statement in the “Confidential to Editor” section, and submit your "Accept" recommendation.

Reviewer #1: All comments have been addressed

Reviewer #2: All comments have been addressed

2. Is the manuscript technically sound, and do the data support the conclusions?

Reviewer #1: Yes

Reviewer #2: Yes

3. Has the statistical analysis been performed appropriately and rigorously? 

Reviewer #1: Yes

Reviewer #2: Yes

4. Have the authors made all data underlying the findings in their manuscript fully available?

Reviewer #1: Yes

Reviewer #2: Yes

5. Is the manuscript presented in an intelligible fashion and written in standard English?

Reviewer #1: Yes

Reviewer #2: Yes

6. Review Comments to the Author

Reviewer #1: The paper " A retrospective study investigating requests for self-citation during open peer review in a general medicine journal can be accepted now.

Reviewer #2: (No Response)

7. PLOS authors have the option to publish the peer review history of their article (what does this mean?). If published, this will include your full peer review and any attached files.

Reviewer #1: No

Reviewer #2: **Yes: **Nataša Kovačić

---

## [Editor Report · Acceptance letter]

6 Aug 2020

PONE-D-20-13025R1 

A retrospective study investigating requests for self-citation during open peer review in a general medicine journal 

Dear Dr. Hesp:

I'm pleased to inform you that your manuscript has been deemed suitable for publication in PLOS ONE. Congratulations! Your manuscript is now with our production department. 

Kind regards, 

on behalf of

Dr. Sergio A. Useche 

Academic Editor

PLOS ONE